# IL-6 Polymorphism as a Predisposing Genetic Factor for Gestational Diabetes or Preeclampsia Development in Pregnancy with Obesity in Relation to VEGF and VEGFF Receptor Gene Expression Modalities

**DOI:** 10.3390/diagnostics14111206

**Published:** 2024-06-06

**Authors:** Panagiotis Halvatsiotis, Theodora Tsokaki, Vasileios Tsitsis, Lina Palaiodimou, Georgios Tsivgoulis, Iraklis Tsangaris, Maria Ourania Panagiotou, Dimitra Houhoula

**Affiliations:** 12nd Department of Propaedeutic Internal Medicine, National and Kapodistrian University of Athens, University General Hospital “Attikon”, 124 62 Athens, Greece; 2Obstetrics and Gynecology Department, General Hospital of Pyrgos, 271 00 Pyrgos, Greece; 32nd Department of Neurology, National and Kapodistrian University of Athens, University General Hospital “Attikon”, 124 62 Athens, Greece; 42nd Department of Critical Care, National and Kapodistrian University of Athens, University General Hospital “Attikon”, 124 62 Athens, Greece; 5Department of Food Science and Technology, University of West Attica, 122 43 Athens, Greece

**Keywords:** pregnancy, obesity, gestational diabetes, preeclampsia, interleukin 6, VEGF, VEGF receptor, type 2 diabetes mellitus

## Abstract

The increased prevalence of obesity worldwide has been implicated in the alarming rise of the incidence of gestational diabetes and preeclampsia, which are both considered threatening conditions for both mother and fetus. We studied gene polymorphisms of the proinflammatory cytokine Interleukin 6 (IL-6) and the gene expression levels of VEGF (vascular endothelial growth factor) and VEGF-R (endothelial growth factor receptor), all known to be involved in pregnancy complications, aiming to identify possible predisposing risk factors in pregnancies with obesity. The G allele of IL-6 was found to correspond with an increased risk for gestational diabetes and preeclampsia occurrence. Furthermore, in obese pregnant mothers with either gestational diabetes or pre-existing type 2 diabetes and those who developed preeclampsia, it was confirmed that gene expression levels of VEGF were reduced while they were increased for VEGF receptors. We conclude that the genetic profile of an obese pregnant woman shares a common background with that of a patient with pre-existing type 2 diabetes mellitus, and therefore predisposes them to complications in pregnancy.

## 1. Introduction

Nowadays, about one-third of all reproductive-aged women are affected by obesity, and the increasing prevalence of obesity seems to have a negative impact on the gestational period, leading more and more pregnant patients to develop complications such as gestational diabetes (GDM) or preeclampsia (PE), a serious hypertensive disorder of pregnancy [1,2].

Women with GDM bear elevated IL-6 levels [3], implicated in various adverse conditions during pregnancy. Many studies have attempted to elucidate the role of these elevated circulating levels, combined with the presence of various gene polymorphisms, aiming to elucidate predisposing genetic factors for complications of development in high-risk pregnancies. In obese pregnancies, protein expression of IL-6 is also raised [4]. It must be clarified whether this is implicated in the inflammation generation of maternal obesity and subsequently leads to insulin resistance, which is considered the causal and responsible mechanism for adverse outcomes during pregnancy. Certain markers with elevated levels are associated with maternal inflammation and have also been identified in preeclampsia [5,6]. Interleukin-6 involvement has been strongly suspected in various adverse conditions during pregnancy, and many studies have attempted to elucidate the relation of certain IL-6 gene polymorphisms in preeclampsia. However, there is currently not enough information to define inflammatory factors, such as interleukin-6, as prognostic tools for investigating pregnancies with either pre-existing type 2 diabetes or obesity.

The pathophysiology of preeclampsia involves both maternal and fetal/placental factors. Abnormalities in the development of placental vasculature early in pregnancy will lead to placental hypoxia and ischemia, which subsequently generate the release of anti-angiogenic factors into the maternal circulation that alter maternal systemic endothelial function and result in hypertensive syndromes and other manifestations. A variety of pro-angiogenic (VEGF and PIGF-placental growth factor) and anti-angiogenic factors (sFlt-1-soluble fms-like tyrosine kinase-1) influence the developing placenta, with the balance between them considered to be critical for normal placental development. More than 30 different single nucleotide polymorphisms of the VEGF gene have already been identified, and many studies have attempted to evaluate their association with the increased risk and predisposition to preeclampsia. Studies are attempting to determine the expression of VEGF in women with preeclampsia, with a view to determine thoroughly its role in pregnancy complications.

It is reported that the levels of VEGF are higher in women with GDM compared to healthy pregnant women, while they are lower in cases with pre-existing diabetes [7]. Pregnancies complicated by GDM are characterized by increased placental expression of VEGF and it is also independently associated with maternal gestational weight gain, defining the concept of “placental diabesity” [8]. It has also been demonstrated that there is a related dysregulation of VEGF placental receptor expression in GDM and even more in pregnancies with a minor degree of glucose intolerance [8]. Changes in VEGF and its receptor influence the placenta, resulting in abnormal blood vessel proliferation and chronic hypoxia. It is strongly suggested that if VEGF and its corresponding receptors can be balanced at an early stage, the risk of pregnancy complications can be diminished [9]. VEGF is critically important for both placental vasculogenesis and angiogenesis throughout gestation, and de-regulation of the VEGF-VEGFR is implicated directly in various diseases. It is still unclear whether women who develop preeclampsia have an underlying vascular pathology, and thus advancing an early detection method by means of the angiokinetic gene expression would be interesting.

In the current study, we investigated the genetic association of IL-6, which is a pro-inflammatory marker, in pregnant women with either pre-pregnancy obesity or type 2 diabetes and in preeclampsia cases. We also assessed the gene expression of the angiogenic markers VEGF and VEGF receptor gene expression during pregnancy mediated by insulin resistance and abnormal carbohydrate metabolism. Our aim is to support our hypothesis that the insulin resistance state that is present in both pre-existing diabetes and obesity exerts a major pathophysiological role in adverse outcomes during gestation, such as preeclampsia.

## 2. Materials and Methods

### 2.1. Subjects

We recruited 36 women, with a mean age of 32.4 years and all of European descent, to participate in the study. They were divided into 4 study groups (Table 1). The preeclampsia study group had 8 volunteers, while 7 mothers with pre-existing type 2 diabetes participated in the DM2 group, 11 pregnant women who developed gestational diabetes participated in the GDM group, and 10 mothers who did not have any problems in pregnancy served as the control group. The body weight and body mass index (BMI) of participants before and after pregnancy are given in Table 2, both overall and by group. In all women, the mean weight before pregnancy was 78.7 (±17.5 kg), and after pregnancy was 89.4 (±16.7 kg). The mean weight difference was 10.8 (±4.1 kg). Accordingly, the mean BMI of participants before pregnancy was 29 (±5.9 kg/m^2^) and after pregnancy was 32.9 (±5.5 kg/m^2^). No significant differences were observed between the groups regarding body weight or body mass index before or after pregnancy, or in their weight difference during pregnancy. Gestational age differed significantly between groups, with women with preeclampsia having significantly lower gestational age (after Bonferroni correction) compared to women with gestational diabetes (*p* = 0.042) and women in the normal group (*p* = 0.027). Half of the sample were smokers. Also, 22.2% of the sample had been diagnosed with diabetes type 2, and 41.7% with gestational diabetes. Of those diagnosed with type 2 diabetes, 66.6% had been diagnosed in the last 5 years. All women had given their signed consent for participation in the study prior to their involvement.

### 2.2. Measurements

Muscle tissue biopsies were performed in the rectus abdominis muscles in the case of cesarean section and from the perineum muscles when giving birth with normal vaginal delivery. All collected samples were then transported in sterile tubes and stored at −70 °C immediately after sampling.

### 2.3. Genomic DNA Extraction

DNA was directly extracted from muscle tissue specimens using an automatic extractor with the Whole Blood Nucleic Acid Extraction Kit (ZYBIO Company, Chongqing, China) following the protocol recommended. The purity and the quantity of extracted DNA were evaluated spectrophotometrically by calculating the OD260/OD280 (spectrophotometer Epoch, Biotek, Winooski, VT, USA).

### 2.4. Genetic Analysis Restriction Fragment Length Polymorphism (RFLP)-IL-6 Gene Analysis

The restriction enzymes used were as follows:1.The *SfaNI* polymorphism is due to a replacement of G by C at position 174, and primers were designed to amplify the promoter region of the IL-6 gene. The primers used in the PCR were as follows: forward -5′- TGACTTCAGCTTTACTCTTTGT -3′and reverse -5′- CTGATTGGAAACCTTATTAAG-3′. PCR was performed in 50 μL final volume solution using the Master Mix (Hot Start Promega, Madison, WI, USA). The amplification was conducted by a thermal cycler (96-Well Thermal Cycler, Applied Biosystems, Singapore), as follows: an initial denaturation of 10 min at 94 °C, and a final extension of 10 min at 72 °C. The cycle program consisted of a 1 min denaturation at 94 °C, a 1 min, 35 s annealing at 55 °C, and a 1 min extension at 72 °C. PCR products were digested with SfaNI restriction enzyme (New England BioLabs, Beverly, MA, USA) at 37 °C overnight and electrophoresed on a 2% agarose gel. SfaNI RFLP was detected by ethidium bromide staining. The identified genotypes were named according to the presence or absence of the enzyme restriction sites, so SfaNI (G/G), (G/C), and (C/C) are homozygotes for the presence of the site (140/58 bp), heterozygotes for the presence and absence of the site (198/140/58 bp), and homozygotes for the absence of the site (198 bp), respectively.2.*NlaIII* at restriction enzyme (New England BioLabs, Beverly, MA, USA) at 37 °C for 15 min (1 μL *NlaIII*, 5 μL buffer 36 μLH2O, and 8 μL PCR product). The enzyme results in the cutting of the 198 bp amplicon into fragments with a size of 122, 45, and 31 bp, which indicates the presence of a wild-type homozygous CC genotype. In addition, two 167 bp and 31 bp fragments indicated the presence of a homozygous GG genotype, and four fragments of 167, 122, 45, and 31 bp indicated the presence of a heterozygous CG genotype.

### 2.5. RNA Isolation and cDNA Synthesis by Reverse Transcription PCR (RT-PCR)

RNA extraction was performed using the commercial kit Nucleospin®RNA Plus (Macherey-Nagel, Düren, Germany). cDNA was synthesized using Luna Script® RT SuperMix Kit (New England BioLabs, Beverly, MA, USA). A 5 μL aliquot of purified RNA was added to 4 μL of the Luna Script® RT SuperMix Kit. The reaction was performed in a total volume of 20 μL. The RT-PCR was performed by combining cDNA, Platinum Blue PCR Super Mix (Invitrogen, Paisley, UK), and the forward and reverse primers for *VEGF* and *VEGFR*. The forward and reverse primer sequences used are shown in the following sequence table. *Beta actin* (*β-actin*), a housekeeping gene, was used as an internal control. Relative quantification of gene expression was calculated using the ΔΔCT method. Analysis of relative gene expression data was performed using quantitative real-time PCR and method 2 (−Delta Delta C(T)) The reaction was continued for 40 cycles under the following thermal conditions: 95 °C for 5 min (initial denaturation), followed by 40 cycles of 45 s at 95 °C (denaturation), 45 s at 60 °C (annealing), and 45 s at 72 °C (extention). Negative controls (nontemplate water instead of cDNA) were also included to ensure a lack of reagent DNA contamination.

### 2.6. Sequence Table

The sequences, annealing temperatures, and product sizes of the primers used to amplify genes of interest.
**Primer****Primer Sequence (5’-3’)****Annealing Temperature****Product Size (bp)**β-actinF: CAAGATCATTGCTCCTCCTG60 °C90 bpβ-actinR: ATCCACATCTGCTGGAAGGVEGFF: TGCAGATTATGCGGATCAAACC60 °C81 bpVEGFR: TGCATTCACATTTGTTGTGCTGTAGVEGFR1F: CAGGCCCAGTTTCTGCCATT60 °C82 bpVEGFR1R: TTCCAGCTCAGCGTGGTCGTA

### 2.7. Biomedical Ethics Issues

The collection of clinical data was correlated with the laboratory research results and was conducted in such a way as to fully guarantee the patients’ anonymity and personal data confidentiality.

### 2.8. Statistical Analysis

Quantitative variables were tested for normality of distribution with the Kolmogorov–Smirnov test. Variables with a normal distribution are expressed as mean values and standard deviations (Standard Deviation = SD), while those following a normal distribution are expressed as median values and interquartile ranges (interquartile range). Absolute (N) and relative (%) frequencies were used to describe qualitative variables. A comparison of proportions was performed with Fisher’s exact test where appropriate. For the comparison of quantitative variables between more than two groups, the parametric test of analysis of variance (ANOVA) or the non-parametric Kruskal–Wallis test was used. Bonferroni correction was used in the case of multiple testing to control for type I errors. Testing of the relationship between pairs of quantitative variables was assessed with Spearman’s correlation coefficient (rho). Significance levels were two-sided and statistical significance was set at 0.05. The statistical program SPSS 22.0 was used for the analysis. Polymerase chain reaction (real-time PCR), agarose gel electrophoresis, and restriction fragment length polymorphism (RFLP) were used to process the samples.

## 3. Results

We investigated the occurrence rates of the GG, GC, and CC genotypes of the -174 G/C polymorphism of the IL-6 gene in the muscle tissue of the volunteers. The majority (80.6%) of the women were carrying the GG genotype (Table 2). Women with preeclampsia and type 2 diabetes or gestational diabetes predominantly expressed the GG genotype. In contrast, the occurrence rate of the CC genotype was significantly higher in the control group with a normal pregnancy (50%).

VEGF and VEGF-R gene expression were quantified and compared between the groups of women participating in the study. This analysis was performed on the women’s muscle tissue, obtained during childbirth. Expression of both the VEGF and the VEGF-R genes differed significantly between the four study groups (Table 3). According to the pairwise comparison of VEGF expression in the study groups of women, after Bonferroni correction, its levels were significantly lower in women with preeclampsia compared to women with type 2 diabetes mellitus (*p* = 0.001), gestational diabetes mellitus (*p* < 0.001), and women with a normal pregnancy (*p* < 0.001). There was no significant difference in VEGF expression between the rest of the groups.

VEGF values were significantly greater in cases with CC polymorphism, while VEGF-R values were significantly greater in cases with GG polymorphism (Table 4).

### 3.1. Association of the -174 G/C Polymorphism of the IL-6 Gene with Patients’ Characteristics

Table 5 shows the demographic data, the body weight characteristics, as well as the gestational age of the women in the study, in relation to the presence of the GG and CC genotype of the -174 G/C interleukin-6 polymorphism. Women with the GG genotype had significantly higher weight and body mass index both before and after pregnancy. However, their body weight change during pregnancy was significantly less compared to women with the CC genotype (*p* = 0.008). A marginally statistically significant difference between the two groups of women was also recorded regarding gestational age at delivery, which is not evaluated as clinically significant.

### 3.2. Association of VEGF and VEGF-R Gene Expression with Patients’ Characteristics

Regarding demographic and body weight characteristics, VEGF expression was negatively associated with women’s pre-pregnancy weight, while VEGF-R expression was positively associated with women’s pre- and post-pregnancy weight, as well as with gestational weight change (Table 6). Thus, it appears that higher weight before pregnancy was associated with significantly lower expression of VEGF and higher expression of VEGF-R. Additionally, higher post-pregnancy weight was associated with higher VEGF-R expression, while the less weight women gained during pregnancy, the higher the VEGF-R expression. Also, an older gestational age was associated with significantly higher expression of VEGF and significantly lower expression of VEGF-R. Gene expression was not associated with women’s BMI before or after pregnancy.

## 4. Discussion

The sample of our study consisted of 36 pregnant women who were divided into four groups (8 women who developed preeclampsia, 7 who suffered from pre-existing type 2 diabetes mellitus, 11 who developed gestational diabetes, and 10 who did not have any problems during pregnancy and served as the control group). The analysis of the gene expression of VEGF and VEGF-R in the groups of women led to the following findings: consistent with the available literature data, VEGF and VEGF-R expression is inversely proportional. Most studies agree that in pathological conditions, such as pregnancies complicated by preeclampsia or type 2 diabetes mellitus, low expression of VEGF and high expression of the VEGF-R are observed [10]. However, even though a substantial number of studies find increased total VEGF levels in preeclampsia, which may also be related to preeclampsia severity, others report them as reduced or at the same levels as healthy women [11,12,13,14,15,16]. A recent study showed decreased levels of VEGF and increased levels of sFlt-1, the endogenous antagonist of VEGF, in the serum of women with preeclampsia relative to women with gestational hypertension, which were also related to the severity of preeclampsia [17]. Earlier, Levine et al. also reported increased expression of sFlt-1 and a parallel decrease in free VEGF expression in preeclampsia before the onset of maternal symptoms [18]. Lower levels of free VEGF in the serum of women with severe preeclampsia compared to women with mild preeclampsia or without hypertension were also confirmed more recently by Cheng et al. [19]. These findings support the theory that increased placental sFlt-1 production causes endothelial dysfunction through competition with VEGF, which in turn contributes to preeclampsia. In addition, the results for VEGF expression in the systemic circulation are significantly affected by a few methodological parameters, including the different VEGF detection and quantification techniques used in the studies that detect either free or conjugated VEGF, contributing to the heterogeneity of findings between studies [20]. The literature data on the expression of the VEGF-R in complicated pregnancies are scarce and indicate that preeclampsia is associated with increased levels of the receptor [21]. Likewise, our study supports the argument that the expression of VEGF is lower while the expression of VEGF-R is higher, mainly in pregnant women with preeclampsia, but also in those with preexisting diabetes mellitus, compared to women with a normal pregnancy. It should be noted that no statistically significant difference is found between women with gestational diabetes treated with insulin or diet alone. The above data support the theory that the VEGF factor plays an important role in normal angiogenesis in the placenta, while in conditions of hypoxia and hypoperfusion, such as in preeclampsia, its production decreases and VEGF-R levels increase reactively. There is also a huge effect of obesity, with women who are not obese having very high expression of the VEGF factor and very low expression of the VEGF-R factor. A correspondingly significant difference in the expression of the factors also results from the comparison between pregnant women with a normal BMI and those with a high BMI, with higher levels of VEGF-R expression in obese women that approach the levels of VEGF-R in women with gestational diabetes mellitus. These results lead to the conclusion that obesity in pregnancy reflects a state of disturbed angiogenesis and that the genetic profile of the obese pregnant woman refers to that of the diabetic pregnant woman, creating a background for complications in pregnancy.

Interesting findings also arise from the analysis for the presence of the -174 G/C polymorphism of the IL-6 gene in the study sample. Regarding the presence of polymorphism in muscle tissue, the most frequent genotype is GG in all groups of women, except for women with normal pregnancy and normal BMI, who express the CC genotype at a rate of 100%. This reinforces the theory that the C allele probably functions protectively, while on the contrary the G allele (and especially in its homozygous form) is associated with the occurrence of problems in pregnancy. It is worth noting that obesity, even in women who complete a normal pregnancy, seems to be associated with an unfavorable gene profile of IL-6 with an increased frequency of the G allele, which is found in women who experience problems in pregnancy, as we found in our study. As early as 2011, a meta-analysis sought to investigate the role of the inflammatory factors TNF-a, IL-6, and IL-10, assessing both their serum expression levels and the presence of polymorphisms in women with preeclampsia. Only four studies with total data from 396 women with preeclampsia and 507 controls were included in the final analysis for the -174 G/C polymorphism of IL-6. According to the results of the analysis, no correlation was recorded between the polymorphism and preeclampsia, which was expected since all studies available up to that time had negative findings. However, the analysis showed increased levels of IL-6 in women with preeclampsia compared to healthy women. It is worth mentioning that the data available until then for the expression of IL-6 were considerably more than the corresponding data for the presence of polymorphisms [22]. It is also worth mentioning that the findings in our study are in accord with Pacheco-Romero et al. 2021 [23], who showed that the GG genotype and the G allele appeared to be associated with the risk for preeclampsia. Also, the study by Sowmya et al. 2015 [24] showed a strong association of the polymorphism with preeclampsia and suggested that it may be an important genetic regulator in the etiology of premature preeclampsia. The meta-analysis by Veisian et al. 2020 [25] showed that the interleukin-6 -174 G/C polymorphism was associated with an increased risk for preeclampsia in Asians, but not in Caucasians and mixed populations. This observation may reflect differences in the genetic profile of the populations. However, data for Caucasians are too limited to draw reliable conclusions. Also, there are no previous data for the Greek population, as this is the first study.

Previous studies have demonstrated a correlation between IL6 and VEGF. Yu Han Huang etc. proved that IL6 induces vascular endothelial growth factor-C expression in lymphatic endothelial cells, as the effort to understand the underlying mechanism may help in developing novel therapeutic strategies to reduce lymphangiogenesis and tumor metastasis [26]. Another study of Toshiyuki Seki etc. highlighted the role of interleukin 6 as an enhancer of anti-angiogenic therapy for ovarian clear cell carcinoma, as IL-6 induced VEGF production by tumor cells and consequently activated angiogenesis [27]. However, this is the first study that correlates IL6 and VEGF in pregnancies complicated by obesity, diabetes, and preeclampsia. Furthermore, serum Il-6 levels have been suggested to be considered as a diagnostic biomarker for gestational diabetes mellitus [28].

The results of the study are consistent with the theory that complicated pregnancies are characterized by a disturbance in the balance between pro-angiogenic and anti-angiogenic factors, and by an inflammatory environment resulting in the induction of systemic endothelial dysfunction with possible adverse effects. Genetic modifications, either in the presence of inherited polymorphisms or at the level of gene expression, may be potential risk factors or predisposing factors for the occurrence of preeclampsia and pregnancy disorders.

Given the relatively high incidence of gestational diabetes mellitus and preeclampsia in the general population and the potential adverse consequences on pregnancy outcome, elucidating the underlying mechanisms and identifying potential biomarkers for increased risk of pregnancy disorders is of paramount importance and warrants further research.

As with most similar studies conducted on women with pregnancy complications, the present study has as its main limitation the relatively small sample size. However, it is important to note that the study was conducted in a homogeneous population of European descent, a characteristic that is a key advantage in genetic studies. In terms of molecular targets, less explored areas will allow the most understanding of preeclampsia and may further contribute to the development of novel targeted therapeutics or early diagnostic markers.

## 5. Conclusions

The present study supports the hypothesis that the vascular endothelial factor VEGF plays an important role in normal angiogenesis during pregnancy, as its expression is reduced, with a simultaneous increase in the expression of the VEGF-R gene, in complicated pregnancies such as preeclampsia and in pregnancies with pre-existing diabetes mellitus type 2. This observation is consistent with the fact that both conditions are characterized by increased insulin resistance and share a common hemodynamic profile. Furthermore, similar findings are observed in pregnant women with an increased body mass index, leading to the conclusion that the genetic profile of obese pregnant women has a common background with that of the diabetic ones.

As for the analysis of interleukin-6 polymorphisms, it appears that the G allele is associated with the occurrence of pregnancy complications, either in the form of diabetes mellitus or in the form of preeclampsia. It is interesting to observe that obese pregnant women also carry the same polymorphism, supporting its possible contribution to the occurrence of adverse outcomes during gestation such as preeclampsia. Further studies with greater participation are needed, targeted to discover an early promising avenue for predicting the future development of preeclampsia.

## Figures and Tables

**Table 1 diagnostics-14-01206-t001:** Sample characteristics in total sample and by group.

	Total Sample (*N* = 36)	Group				*p*
		Preeclampsia (*N* = 8; 22.2%)	Diabetes Type 2 (*N* = 7; 19.4%)	Gestational Diabetes (Diet or Insulin) (*N* = 11; 30.6%)	Normal (*N* = 10; 27.8%)	
	Mean (SD)	Mean (SD)	Mean (SD)	Mean (SD)	Mean (SD)	
Age (years)	32.4 (4.6)	33.8 (4.2)	30.9 (4.4)	33.3 (3.6)	31.5 (5.9)	0.537 ‡
Weight before pregnancy (kg/m^2^)	78.7 (17.5)	81 (17.5)	88.6 (18.1)	78.2 (19.9)	70.6 (11.5)	0.209 ‡
Weight after pregnancy (kg/m^2^)	89.4 (16.7)	91 (16.9)	98.7 (17.6)	88.3 (19.6)	82.7 (10.2)	0.277 ‡
Weight difference	10.8 (4.1)	10 (2.9)	10.1 (2.6)	10.5 (4.8)	12.2 (5.2)	0.685 ‡
BMI before pregnancy (kg/m^2^)	29.0 (5.9)	29.0 (6.7)	32.9 (6.9)	28.5 (5.8)	26.7 (3.9)	0.204 ‡
BMI after pregnancy (kg/m^2^)	32.9 (5.5)	32.6 (6.2)	36.7 (6.4)	32.3 (5.5)	31.3 (3.4)	0.232 ‡
Gestational age (weeks)	38.2 (0.7)	37.6 (0.7)	38.1 (0.7)	38.4 (0.7)	38.5 (0.6)	0.020 ‡
	*n* (%)	*n*	*n*	*n*	*n*	
Smoking	18 (50.0)	4	3	5	6	0.937 +
Diabetes						
No	13 (36.1)	3	0	0	10	<0.001 +
Yes, during pregnancy	15 (41.7)	4	0	11	0	
Yes, type 2	8 (22.2)	1	7	0	0	
Years from type 2 diabetes diagnosis						
<1 year	2 (33.3)	0	2	0	0	0.333 +
1–5 years	2 (33.3)	0	2	0	0	
5–10 years	1 (16.7)	0	1	0	0	
>10 years	1 (16.7)	1	0	0	0	

‡ ANOVA; + Fisher’s exact test.

**Table 2 diagnostics-14-01206-t002:** Absolute and relative frequencies of occurrence of GG, GC, and CC genotypes of the -174 G/C polymorphism of the IL-6 gene in all 4 groups of women in the study.

Polymorphism IL-6 -174 G/C	Total Sample (*N* = 36)		Group				*p* Fisher’s Exact Test
			Pre-ecl Ampsia (*N* = 8; 22.2%)	Diabetes Type 2 (*N* = 7; 19.4%)	Gestational Diabetes (Diet or Insulin) (*N* = 11; 30.6%)	Normal (*N* = 10; 27.8%)	
	*N*	%	*N*	*N*	*N*	*N*	
CC	6	16.7	0	1	0	5	**0.009**
GC	1	2.8	0	0	1	0	
GG	29	80.6	8	6	10	5	

**Table 3 diagnostics-14-01206-t003:** VEGF and VEGF-R gene expression levels in muscle tissue, in all 4 groups of women in the study.

	Total Sample (*N* = 36)		Group								*p* Kruskal–Wallis Test
			Preecl Ampsia (*N* = 8; 22.2%)		Diabetes Type 2 (*N* = 7; 19.4%)		Gestational Diabetes (Diet or Insulin) (*N* = 11; 30.6%)		Normal (*N* = 10; 27.8%)		
	Mean (SD)	Median (IQR)	Mean (SD)	Median (IQR)	Mean (SD)	Median (IQR)	Mean (SD)	Median (IQR)	Mean (SD)	Median (IQR)	
VEGF	10.2 (2.3)	10.7 (8.7–11.4)	7.1 (0.3)	7.2 (7–7.3)	9.5 (1.2)	8.8 (8.7–11)	11.2 (0.3)	11 (10.9–11.5)	12.2 (2.2)	11.2 (10.5–15.1)	**<0.001**
VEGF -R	12.0 (2.2)	12.3 (11.3–13.6)	14.4 (0.4)	14.3 (14.1–14.9)	12.6 (1.2)	13.1 (11.3–13.4)	11.6 (0.4)	11.4 (11.3–12.1)	10.1 (2.7)	11.2 (6.5–12.3)	**<0.001**

**Table 4 diagnostics-14-01206-t004:** VEGF and VEGF-R gene expression levels in muscle tissue according to Polymorphism IL-6 -174 G/.

	Polymorphism IL-6 -174 G/C				*p* Mann-Whitney Test
	CC		GG		
	Mean (SD)	Median (IQR)	Mean (SD)	Median (IQR)	
VEGF	13.2 (2.3)	13.6 (11–15.3)	9.6 (1.8)	10.4 (7.6–11)	0.005
VEGF-R	8.7 (2.6)	8.3 (6.4–10.5)	12.7 (1.4)	12.4 (11.4–14)	0.002

Note. Category CG was not analyzed due to small sample size (*N* = 1).

**Table 5 diagnostics-14-01206-t005:** Association of the -174 G/C polymorphism of the IL-6 gene with demographic and body weight characteristics of women in the study.

	Polymorphism				*p* Student’s *t*-Test
	CC		GG		
	Mean	SD	Mean	SD	
Age	31.0	5.9	32.6	4.3	0.459
Weight before pregnancy	61.3	6.3	81.9	17.1	**0.007**
Weight after pregnancy	76.1	9.6	91.7	16.8	**0.035**
Weight difference	14.8	4.3	9.9	3.7	**0.008**
Body mass index before pregnancy	22.9	0.8	30.1	5.8	**0.005**
Body mass index after pregnancy	28.4	2.2	33.7	5.6	**0.029**
Gestational age in weeks at delivery	38.7	0.6	38.1	0.7	**0.046**

**Table 6 diagnostics-14-01206-t006:** Correlation of VEGF gene and VEGF-R receptor expression levels with characteristics of women in the study.

	VEGF		VEGF-R	
	rho	*p*	rho	*p*
Age	−0.04	0.823	0.06	0.729
Weight before pregnancy	−0.36	**0.029**	0.41	**0.012**
Weight after pregnancy	−0.30	0.071	0.36	**0.033**
Weight difference	0.32	0.056	−0.35	**0.039**
Body mass index before pregnancy	−0.25	0.136	0.31	0.070
Body mass index after pregnancy	−0.27	0.106	0.32	0.057
Gestational age in weeks at delivery	0.54	**0.001**	−0.56	**<0.001**

## Data Availability

All data are available upon request.

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
