# Peer review of "IL-6 Polymorphism as a Predisposing Genetic Factor for Gestational Diabetes or Preeclampsia Development in Pregnancy with Obesity in Relation to VEGF and VEGFF Receptor Gene Expression Modalities"

_diagnostics, 2024, doi:10.3390/diagnostics14111206_

Round 1

Reviewer 1 Report

Comments and Suggestions for Authors

Esteemed Editor and author team,

The study is interesting and can be published after a major revision.

There are several problems:

1. The small number of cases raises doubts about the value of statistical calculations.

2. The results for IL-6 and respective VEGF/VEGF-R are not correlated. The authors should try to find statistical correlations.

3. The references are insufficient. I propose that they be supplemented with other bibliographic titles and, consequently, the discussion chapter be redone.

Ex: https://www.ncbi.nlm.nih.gov/pmc/articles/PMC7393755/

4. The description of the study group is inadequate.

Ex :

In the chapter "Discussions" it is stated that the study group is homogeneous / Caucasian. This aspect must also be found in the chapter "Material and method".

5. It is necessary to restore the tables by changing the graphics.

Best regards

Author Response

We sincerely thank you for your constructive comments related to our manuscript entitled “IL-6 polymorphism as a predisposing genetic factor for gestational diabetes or preeclampsia development in pregnancy with obesity in relation to VEGF and VEGFF receptor gene expression modalities.”

We have addressed all the reviewer’s issues and comments below and we hope that it is now suitable to fulfill the journal’s requirements.

Again, thank you very much for your attention to this manuscript.

The study is interesting and can be published after a major revision.

There are several problems:

  1. The small number of cases raises doubts about the value of statistical calculations.

We agree that the number of volunteers is small, and we have stated that is the major limitation of our study. However, the statistical evaluation demonstrated various strong correlations that might assist in the design of any further investigations in the field.

  1. The results for IL-6 and respective VEGF/VEGF-R are not correlated. The authors should try to find statistical correlations.

Thank you for your comment. An association between IL-6 and VEGF/VEGF-R was added in the revised manuscript.

  1. The references are insufficient. I propose that they be supplemented with other bibliographic titles and, consequently, the discussion chapter be redone.

Ex: https://www.ncbi.nlm.nih.gov/pmc/articles/PMC7393755/

We agree and we have supplemented further bibliographic titles and we enhanced the Discussion section

  1. The description of the study group is inadequate.

Ex : In the chapter "Discussions" it is stated that the study group is homogeneous / Caucasian. This aspect must also be found in the chapter "Material and method".

We agree and we have included all the relevant demographic information in Material and method section.

  1. It is necessary to restore the tables by changing the graphics.

We are also included in the revised manuscript your suggestions.

Reviewer 2 Report

Comments and Suggestions for Authors

The authors present the results of the study aimed to evaluate IL-6 polymorphisms and VEGF/VEGFR expression in women with preeclampsia/T2DM/GDM. The research is very timely due to the importance of the studied disorders. However, manuscript contains the certain issues to be addressed:

1.       The title requires correction, as the article does not present the results interconnecting IL-6 polymorphisms and VEGF/VEGFR expression

2.       Interconnection between IL-6 and VEGF is not displayed in the introduction and the study of these molecules in combination is not justified. Meanwhile, there may be found relevant studies, the data in which are not that very straightforward: https://www.ncbi.nlm.nih.gov/pmc/articles/PMC4934912/

https://www.nature.com/articles/s41598-021-86913-9

3.       Why skeletal muscle tissue was chosen for analysis? Especially for IL-6 polymorphisms, which will be the same in all the cells, including PBMC, which are easier to obtain? Please, explain.

4.       Basic clinical characteristics of patients (age, BMI, duration of DM2 if applicable, etc.) are absent.

5.       Pairwise comparison of individual groups should have been performed by the Mann–Whitney U test with Bonferroni correction in  case of the multiple comparisons

6.       If n<20 the frequencies should not be presented in %, only n, as % will give incorrect impression.

7.       May be, with such a low number of patients being recruited at this moment, it is better not to identify 6 groups of patients in addition to division of patients into 4 groups. Increase of the number of groups to compare > 2 requires application of Bonferroni correction in pairwise comparison: with n=6, the p-level should be multiplied by 6. In this case in many cases the p-level will be above 0.05 (VEGF expression in women with T2DM vs women with gestational DM (0.027*6 =0.162 ); etc. )  and differences appear to be insignificant.

8.       In this case the conclusion in the abstract should be changed.

9.       Figures duplicate tables – this should be corrected.

10.   Table headings should be above tables, unlike figure legends

11.   In discussion: Likewise, our study supports that the expression of VEGF is higher, while that of VEGF-R is lower mainly in pregnant women with preeclampsia but also in those with preexisting Diabetes Mellitus, compared to women with a normal pregnancy  - according to the results, VEGF was lower, while VEGF-R was higher.

12.   The conclusions: The present study supports the hypothesis that the vascular endothelial factor VEGF plays an important role for normal placental development … - the placenta development was not evaluated in this study. Please, rewrite this part. + It is interesting to observe that obese pregnant women also carry the same polymorphism, supporting its possible contri-bution to the occurrence of problems during pregnancy – “problems during pregnancy” – sounds too unspecific. Please, try to formulate more correctly.

13.   Minor:

-          “VEGF factor” and “VEGFR receptor” in discussion are incorrect – F in abbreviation stands for “factor”, and in VEGFR – R stands for “receptor”

-          Please, decipher VEGF, PlGF, sFlt-1 in introduction after the first appearance

Comments on the Quality of English Language

Some sentences need to be rewritten:

1. However, does not exist till today enough information in order various inflammatory markers, such as IL-6 to serve as predictive tools, because we are lacking studies investigating pregnancies with either preexisting type 2 diabetes or obesity. (the structure of the sentence is incorrect)

2. Changes in VEGF and its receptor act (influence the placenta???) on the placenta, resulting in abnormal blood vessel proliferation and chronic hypoxia.

Author Response

The authors present the results of the study aimed to evaluate IL-6 polymorphisms and VEGF/VEGFR expression in women with preeclampsia/T2DM/GDM. The research is very timely due to the importance of the studied disorders. However, manuscript contains the certain issues to be addressed:

  1. The title requires correction, as the article does not present the results interconnecting IL-6 polymorphisms and VEGF/VEGFR expression.

We agree and we added in the manuscript the relevant statistical evaluations for possible correlations in order to better constitute with the title

  1. Interconnection between IL-6 and VEGF is not displayed in the introduction and the study of these molecules in combination is not justified. Meanwhile, there may be found relevant studies, the data in which are not that very straightforward: https://www.ncbi.nlm.nih.gov/pmc/articles/PMC4934912/ https://www.nature.com/articles/s41598-021-86913-9

Thank you for your comment. An association between IL-6 and VEGF/VEGF-R was added in the revised manuscript.

  1. Why skeletal muscle tissue was chosen for analysis? Especially for IL-6 polymorphisms, which will be the same in all the cells, including PBMC, which are easier to obtain? Please, explain.

Skeletal muscle participates significantly in the insulin resistance process and is easily accessible during a cesarean section with a very minor and insignificant intervention for the mothers.  

  1. Basic clinical characteristics of patients (age, BMI, duration of DM2 if applicable, etc.) are absent.

We agree and we have included all the relevant demographic information and a table with participants’ characteristics was added.

  1. Pairwise comparison of individual groups should have been performed by the Mann–Whitney U test with Bonferroni correction in case of the multiple comparisons

Thank you for your comment. The analysis was corrected according to your comment.

  1. If n<20 the frequencies should not be presented in %, only n, as % will give incorrect impression.

In these cases, the percentages were removed.

  1. May be, with such a low number of patients being recruited at this moment, it is better not to identify 6 groups of patients in addition to division of patients into 4 groups. Increase of the number of groups to compare > 2 requires application of Bonferroni correction in pairwise comparison: with n=6, the p-level should be multiplied by 6. In this case in many cases the p-level will be above 0.05 (VEGF expression in women with T2DM vs women with gestational DM (0.027*6 =0.162 ); etc. ) and differences appear to be insignificant.

The analyses referring to the 6 groups were removed.

 8. In this case the conclusion in the abstract should be changed.

      We made the requested corrections.

  1. Figures duplicate tables – this should be corrected.

The figures were removed.

  1. Table headings should be above tables, unlike figure legends

The titles were moved above the tables.

  1. In discussion: Likewise, our study supports that the expression of VEGF is higher, while that of VEGF-R is lower mainly in pregnant women with preeclampsia but also in those with preexisting Diabetes Mellitus, compared to women with a normal pregnancy - according to the results, VEGF was lower, while VEGF-R was higher.

Thank you for your comment. The manuscript was corrected according to your comment.

  1. The conclusions: The present study supports the hypothesis that the vascular endothelial factor VEGF plays an important role for normal placental development … - the placenta development was not evaluated in this study. Please, rewrite this part. + It is interesting to observe that obese pregnant women also carry the same polymorphism, supporting its possible contri-bution to the occurrence of problems during pregnancy – “problems during pregnancy” – sounds too unspecific. Please, try to formulate more correctly.

Thank you for your comment. The manuscript was corrected according to your comment.

  1. Minor:

-          “VEGF factor” and “VEGFR receptor” in discussion are incorrect – F in abbreviation stands for “factor”, and in VEGFR – R stands for “receptor”

-          Please, decipher VEGF, PlGF, sFlt-1 in introduction after the first appearance

Thank you for your comments. The manuscript was corrected according to your comments.

Comments on the Quality of English Language

The revised manuscript was edited by an English language educator.

Some sentences need to be rewritten:

  1. However, does not exist till today enough information in order various inflammatory markers, such as IL-6 to serve as predictive tools, because we are lacking studies investigating pregnancies with either preexisting type 2 diabetes or obesity. (the structure of the sentence is incorrect)
  2. Changes in VEGF and its receptor act (influence the placenta???) on the placenta, resulting in abnormal blood vessel proliferation and chronic hypoxia.

Thank you for your comments. The manuscript was corrected and rephrased according to your comments.

Round 2

Reviewer 1 Report

Comments and Suggestions for Authors

Esteemed Editor and author team,

In my opinion the study can be accepted in its current form for publication.

Best regards

Reviewer 2 Report

Comments and Suggestions for Authors

The introduced changes significantly improved the manuscript.

I have no further suggestions.